# *Zymomonas mobilis* in Bread Dough: Characterization of Dough Leavening Performance in Presence of Sucrose

**DOI:** 10.3390/foods9010089

**Published:** 2020-01-15

**Authors:** Alida Musatti, Carola Cappa, Chiara Mapelli, Cristina Alamprese, Manuela Rollini

**Affiliations:** Dipartimento di Scienze per gli Alimenti, la Nutrizione, l’Ambiente, Università degli Studi di Milano, Via G. Celoria, 2-20133 Milano, Italy

**Keywords:** *Zymomonas mobilis*, leavening properties, yeast-free products, sucrose

## Abstract

*Zymomonas mobilis*, because of its fermentative metabolism, has potential food applications in the development of leavened baked goods consumable by people with adverse responses to *Saccharomyces cerevisiae*. Since *Z. mobilis* is not able to utilize maltose present in flour, the effect of sucrose addition (2.5 g/100 g flour) on bread dough leavening properties was studied. For comparison purposes, leavening performances of *S. cerevisiae* with and without sucrose were also investigated. Doughs leavened by *Z. mobilis* without sucrose addition showed the lowest height development (14.95 ± 0.21 mm) and CO_2_ production (855 ± 136 mL). When sucrose was added, fermentative performances of *Z. mobilis* significantly (*p* < 0.05) improved (+80% and +85% of gas production and retention, respectively), with a dough maximum height 2.6 times higher, results indicating that *Z. mobilis* with sucrose can be leavened in shorter time with respect to the sample without addition. *S. cerevisiae* did not benefit the sucrose addition in terms of CO_2_ production and retention, even if lag leavening time was significantly (*p* < 0.05) shorter (about the half) and time of porosity appearance significantly (*p* < 0.05) longer (about 26%) with respect to *S. cerevisiae* alone. Results demonstrate that in the presence of sucrose, *Z. mobilis* can efficiently leaven a bread dough, thus providing innovation possibilities in the area of yeast-free leavened products.

## 1. Introduction

*Saccharomyces cerevisiae* is by far the most common microorganism used for alcoholic beverage and leavened dough production. Human exposure to this yeast species is massive; not only these products contains *S. cerevisiae*, but also multi-vitamin food supplements, probiotics preparation, and even vaccines production [1]. So far, the possibility of adverse responses related to its ingestion is spreading [2].

While food intolerance is described as an adverse reaction to food not determined by a cell-mediated immune response [3,4], food allergy is described as an exaggerate immune system response against food components [5] with consequent antibodies production [6].

In patients with inflammatory bowel disease or Crohn’s disease, *S. cerevisiae* cell wall components have been recognized as antigens and anti-*S. cerevisiae* antibodies (ASCA) can be used as specific diagnostic markers [7,8,9]. However, investigations on the physiological mechanisms that may contribute to the onset of *S. cerevisiae* allergy and/or intolerance are still scarcely documented within the scientific literature. Indeed, in all these patients, dietary restrictions avoiding the ingestion of foods in which *S. cerevisiae* is present are recommended [10]. 

With the aim of fulfilling the requirement of baked goods consumable by people with adverse responses to the ingestion of *S. cerevisiae*, scientific research has been focusing great attention toward the development of yeast-free leavened products. De Bellis et al. [11] recently investigated the use of *Leuconostoc citreum* as starter for a liquid type-II sourdough. Because of its similarity to *S. cerevisiae* fermentation metabolism, the Gram-negative bacterium *Zymomonas mobilis* can also be considered as an attractive alternative to yeast in dough leavening [12]. *Z. mobilis*, classified as GRAS by FDA [13], possesses a narrow spectrum of substrates employable as carbon and energy source: only sucrose, glucose, and fructose can be metabolized through the Entner-Doudoroff pathway, giving ethanol and CO_2_ as final products [14].

Nevertheless, in wheat flour, the main fermentable sugar is maltose, whose concentration ranges from 1.7 to 3 g/100 g [15], while only 0.22–0.43 g/100 g is the sum of glucose, fructose, and sucrose fermentable by *Z. mobilis* [16]. 

To increase glucose availability, Musatti et al. [17] investigated the possibility of obtaining a gradual glucose release in a model dough exploiting the constitutive maltose hydrolytic activity of *Lactobacillus sanfranciscensis*, an obligate heterofermentative key bacterium that dominates in traditional type I sourdough [18]. The results confirmed that when the two microorganisms were combined at high cell concentration (9 log CFU/g dough), CO_2_ production was higher than the mathematical sum of the gas produced by the single bacteria, but only during the first hours of leavening. The subsequent efficiency loss may be due to several factors, above all glucose shortage for *Z. mobilis*, as well as the decrease of dough pH that can negatively affect both *Lactobacillus* and *Zymomonas* metabolism [17]. 

Another applied strategy to overcome the limited amount of sugars fermentable by the bacterium may focus on the addition to the dough formulation of sugars (e.g., sucrose and glucose) that *Zymomonas* is able to ferment. Oda and Tonomura [12] reported good leavening abilities in presence of 5 g/100 g flour of sucrose, whereas higher amounts (up to 35 g/100 g flour) decrease *Z. mobilis* fermentative performances. Musatti et al. [19] demonstrated that in presence of 1 or 5 g/100 g flour, *Z. mobilis* efficiently leavens a dough; the higher the glucose addition, the higher the CO_2_ produced. However, the highest amount of glucose tested (5 g/100 g flour) was not completely consumed by Z. *mobilis*; this could negatively affect the final taste of the bread, imparting high sweetness and modifying its nutritional properties because of the high glycaemic impact and the amount of Maillard compounds. 

To the best of our knowledge, no data are available concerning sucrose addition lower than 5 g/100 g flour in bread dough and not always the leavening performances of *S. cerevisiae* have been compared in the same experimental conditions. Note also that sucrose is a cheaper carbon source rather than glucose (263 vs. 336 Euro/10 kg, Sigma-Aldrich, St Louis, MO, USA). The aim of this work is to investigate the effect of a low sucrose addition (2.5 g/100 g flour) on the dough leavening performance of *Z. mobilis*, in order to limit any possible modification of the sensory and nutritional features of the final leavened product. For comparison purposes, fermentative performances of the traditional baking yeast were also investigated and used as benchmark, applying the same cell concentration (7 log CFU/g) used for *Z. mobilis*. Doughs were characterized in terms of volume development, CO_2_ production and retention for leavening times up to 24 h. The lag leavening time and leavening rates were also calculated from the reofermentographic curves obtained for each tested condition. Time course of microbial population, sugar consumption, and ethanol production were also analyzed. 

## 2. Materials and Methods

### 2.1. Materials and Samples

A strong Manitoba flour (14 g/100 g proteins; Carrefour, Milan, Italy) was used. Compressed baker’s yeast (Lievital, Lesaffre Italia S.p.A., S. QuiricoTrecasali, Parma, Italy) and sucrose (Carrefour,) were purchased at a local supermarket and yeast was stored at 4 °C until use within the first two weeks after purchase. *Zymomonas mobilis subs. mobilis* type strain DSM 424 (Deutsche Sammlung von Mikroorganismen und Zellkulturen GmbH, Braunschweig, Germany) was used. *Z. mobilis* strain maintenance and biomass production were performed as reported by Musatti et al. [17]. In the case of compressed bakers’ yeast, to calculate the cell amount (log CFU/g) to add to the dough in order to reach the same level of inoculum of *Z. mobilis*, a weighed portion was suspended in 9× physiological solution (NaCl 9 g/L); the obtained mixture was subjected to serial decimal dilutions and plated in Yeast Glucose Chloramphenicol Agar (YGC, Scharlab, Barcelona, Spain), incubated at 25 °C for 3–5 days.

Dough samples were produced and tested in duplicate (Table 1), leavened by *Z. mobilis* or *S. cerevisiae*, without or with sucrose addition (2.5 g/100 g flour). Microbial cell inoculum was maintained constant at a low level (7 log CFU/g dough) similar to a sourdough, being *Z. mobilis* cell production far more expensive than that of *S. cerevisiae* because of the much lower biomass yield [20,21].

### 2.2. Flour Characterization and Dough Production

Flour mixing properties without or with sucrose addition (2.5 g/100 g of flour) were assessed by means of a Brabender^®^ Farinograph (Brabender OHG, Duisburg, Germany; 300 g chamber, 30 °C), a worldwide standard for testing flour quality. Flour (300 g) was pre-mixed for 1 min, then water (sucrose when needed was dissolved in it) was added to the flour up to reach a dough consistency of 500 ± 25 BU (Brabender Unit); water absorption (g/100 g), arrival time (min), dough consistency (BU), and dough stability (min) were measured. Data were reported as mean and standard deviation values of two replicates. Dough samples were produced using the same equipment used for testing flour mixing properties (Brabender^®^ Farinograph) and taking into account the water absorption values of flour with or without sucrose addition. Sucrose was dissolved in water before addition to flour. Microbial biomass was added to flour in liquid form, while compressed yeast was suspended in water immediately before the trial. Kneading was carried out for 8 min at 30 °C in order to ensure a complete hydration of the ingredients and a well-developed protein network. All the samples had a dough consistency of 500 ± 25 BU that guarantees the workability of the dough by hand or an industrial forming machine [22]. Dough sample identifications and formulations are summarized in Table 1. Each dough was produced twice in order to have two technological replicates.

### 2.3. Microbial Population Counts

Immediately after dough production (t0) and after 8 and 24 h (t8 and t24) of leavening, 5–8 g of dough were diluted in 45–72 mL sterile peptone water (10 g/L bacto-peptone in deionized water, pH 6.8) and homogenized in a Stomacher 400 Circulator (Seward, Worthing West Sussex, UK) for 5 min at 260 rpm. After decimal dilutions in the same solution, suspensions were plated in appropriate media: *Z. mobilis* was plated onto DSM agar (DSM broth added of 15 g/L agar), incubating at 30 °C for 3 days in anaerobic conditions; total bacterial count (TBC) was determined by pour-plating in Tryptic Soy Agar (TSA, Scharlab) after incubation at 30 °C for 48–72 h; TSA was added of 0.1% (v/v) Cycloheximide (Sigma-Aldrich) for avoiding *S. cerevisiae* growth. Yeasts were determined by pour-plating in Yeast Glucose Chloramphenicol Agar (YGC, Scharlab) and incubated at 25 °C for 3–5 days. Counts were reported as logarithms of the number of colony forming units (log CFU/g of dough) as mean and standard deviation values of two technological replicates.

### 2.4. Sugars and Ethanol Determination

Maltose, glucose, and sucrose were determined in duplicate immediately after dough production (t0) and after 8 and 24 h of leavening by using the maltose, sucrose, and D-glucose enzymatic kit (K-Masug 11/16, Megazyme, Bray Co. Wicklow, Ireland). Ethanol produced at the same sampling times was determined by HPLC as reported by Musatti et al. [17]. Results represent the average and standard deviation values of two technological replicates and are referred to 100 g of flour (g/100 g).

### 2.5. Dough Leavening Properties

Dough development and CO_2_ production and retention during leavening were continuously recorded at 26 °C for 24 h by a Chopin Rheofermentometer F3 (Chopin, Villeneuve-La-Garenne Cedex, France) as reported by Cappa et al. [23]. The following indices were taken from the resulting curves: dough maximum height (Hm, mm); time corresponding to dough maximum height (T1, h); total gaseous production (CO_2_-TOT, mL); CO_2_ retained (CO_2_-RET, mL); time of dough porosity appearance (Tx, h). Two other unconventional indices were extrapolated from the dough development curves: The lag leavening time (LLT, h), defined as the time before an increase in dough height was noticed; and the leavening rate (LR, mm/h), as the slope of the first linear part of the curve after LLT. Results represent the average and standard deviation values of two technological replicates.

### 2.6. Dough pH Evolution during Leavening

During leavening, pH was continuously recorded on 50 g of each dough sample using a pH-meter PHM 220 (Radiometer, A. De Mori Strumenti S.p.A., Milan, Italy). Results represent the average and standard deviation values of two technological replicates.

### 2.7. Statistical Analysis

Analytical results were treated by one-way analysis of variance (ANOVA), followed by the least significant difference (Fisher’s LSD) test to highlight significant differences (*p* < 0.05) among samples. Data were processed by Statgraphics Centurion (v. 18, Statistical Graphics Corp., Herndon, VA, USA).

## 3. Results and Discussion

### 3.1. Flour Characterization

The flour used in the present study is a type 0 Manitoba, a variety of common wheat (*Triticum aestivum*) characterized by a high protein content. Flour analysis confirmed a high protein content of 12.89 ± 0.08 g/100 g db. The amount of water necessary to reach a dough consistency of 500 ± 25 BU (62.0 ± 1.8 g/100 g of flour), the arrival time (2.0 ± 0.3 min), and the dough stability (12.5 ± 0.5 min) indicated that the dough can withstand long mixing and leavening phases typical of bread and pizza production (Figure 1).

Indeed, the Italian voluntary classification of wheat [24] defines flour having protein content of 11.5–13.4 and 13.5–14.4 g/100 g db and farinographic stability ≥5 and ≥10 min as “bread-making flour” and “superior bread-making wheat flour,” respectively.

When sucrose was added to the flour, no significantly (*p* < 0.05) different water absorption (57.7 ± 0.7 g/100 g of flour) and arrival time (2.0 ± 0.1 min) values were obtained, whereas dough stability (18.0 ± 0.3 min) significantly (*p* < 0.05) increased, indicating that the amount of added sucrose partially affected the flour-mixing properties (Figure 1). 

### 3.2. Sugar, Ethanol, pH, and Microbial Evolution during Dough Leavening

Sugar and ethanol evolutions during leavening are reported in Table 2, while time course of microbial counts and pH are shown in Figure 2 and Figure 3, respectively. In dough leavened by *Z. mobilis* (sample Z), maltose remained unfermented and accumulated up to 3.85 ± 0.08 g/100 g flour after 24 h, because of the hydrolytic action of endogenous amylases [17].

On the contrary, the little amount of glucose naturally present in flour was promptly consumed and converted into ethanol, reaching the maximum value of 0.75 g/100 g flour. *Z. mobilis* grew slightly from 7.5 to 8 log CFU/g while TBC sharply increased more than 4 log cycles, from 4 to 8 log CFU/g, owing to the availability of maltose that can be consumed by other microbial strains rather than *Z. mobilis*. Dough pH slowly decreased from 6 to around 5.4 in 24 h, because of the limited production of organic acids by *Z. mobilis* [14]. In doughs leavened by *S. cerevisiae* (sample S), compressed bakers’ yeast was used within the first 2 weeks of purchase; even if shelf life of compressed yeast usually lies between 35 and 40 days at refrigerated temperature, intracellular glutathione concentration is high at the beginning of shelf life and decreases during storage, being considered an indicator of cell stress tolerance [25]. As expected maltose and glucose were completely consumed at the end of leavening, resulting in a final ethanol concentration of 3.49 g/100 g flour, significantly (*p* < 0.05) higher than in Z dough. During leavening, yeast population increased from 6.8 to around 8.0 log CFU/g after 24 h, while TBC grew less than in Z (only 2 log cycles, from 4 to 6 log CFU/g), because of the limited amount of sugars consumed by *S. cerevisiae* and to the antimicrobial effect of ethanol. Dough pH trend was similar to that of sample Z, because *S. cerevisiae* produces organic acids, mainly succinic, acetic, and carbonic acids only during the respiration phase in the first leavening period; when oxygen is consumed, the organism switches to alcoholic fermentation, organic acid production is inhibited, and pH stabilized [26]. When sucrose was added to the dough fermented by *Z. mobilis* (sample Zs), its concentration started to decrease after 8 h of leavening and was totally consumed at 24 h. From 8 to 24 h, an increase in *Z. mobilis* population was observed and the final ethanol concentration was significantly (*p* < 0.05) higher (1.72 g/100 g) than in Z sample without sucrose addition.

No relevant changes were observed in pH trends with respect to Z sample, while a lower TBC growth was registered from 4 to 6 log CFU/g, because of the presence of ethanol. Dough samples added with sucrose and leavened by *S. cerevisiae* (sample Ss) evidenced a certain amount of fermentable sugars, around 0.5 g/100 g present even after 24 h. Yeast was also found to grow and ferment less (ethanol 2.58 g/100 g) than in sample S, while dough pH remained at higher levels. Such a behavior highlighted that the presence of sucrose produces an osmotic stress able to inhibit *S. cerevisiae* metabolism. Note that *S. cerevisiae* consumed sucrose more rapidly than *Z. mobilis* because of the presence of invertase [26,27]. In *Z. mobilis*, sucrose can be split extracellularly or intracellularly but also levan can be produced via levansucrase [14]. *S. cerevisiae* prefers sucrose to maltose, the latter being transported into the cell with the help of a specific transmembrane transporter before use [28].

### 3.3. Dough Leavening Properties

Dough technological evaluation results are reported in Table 3 and Figure 3. According to the low microbial concentration, dough rising needed long time to start. Sample Z without sucrose addition showed a long lag leavening time, more than double respect to S, confirming the low fermentative activity of *Z. mobilis* in comparison to baker’s yeast. Accordingly, T1 was about 60% longer in Z than in S, and leavening rate was about 65% lower. The majority (86%) of the CO_2_ produced was retained by Z dough, but the lowest dough development was obtained because the CO_2_ TOT was significantly lower (*p* < 0.05) in comparison to the other samples. In a real bread-making process, the dough leavened using 7 log CFU/g of *Z. mobilis* should have optimum leavening length shorter than 10 h, because after this time the dough started to lose CO_2_. 

*S. cerevisiae* did not benefit of sucrose addition in terms of CO_2_ production and retention, but LLT was significantly (*p* < 0.05) shorter (about the half) and Tx significantly (*p* < 0.05) longer, about 26% with respect to S. The dough started to lose CO_2_ after a longer time because the presence of sucrose contributed to increase dough viscous characteristics and extensibility [29], thus improving gas retention capacity [26]. This behavior is also in agreement with the higher farinographic stability of the dough containing sucrose (Figure 1). 

On the contrary, sucrose addition definitely enhanced *Z. mobilis* leavening performances: Zs showed a significant (*p* < 0.05) increase of both CO_2_ TOT and CO_2_ RET, around +80 and +85% respectively, in comparison to sample Z, resulting also in a significantly (*p* < 0.05) higher Hm, about 2.6 times higher. These results are in agreement with the microbial counts and ethanol concentrations measured after 24 h leavening (Figure 2; Table 2). Sample Zs showed also a significantly (*p* < 0.05) lower LLT of about 1 h, a higher LR (+68%) and a Tx postponed of about 30 min, indicating that Zs can be leavened in shorter time with respect to the sample Z. 

The improvement of *Z. mobilis* leavening performances achieved in the present study using 2.5 g/100 g flour is even more interesting considering that *Z. mobilis* is able to completely use the added sugar. This can have clear benefits in terms of final bread taste as well as in nutritional features and color, e.g., lower sweetness and less available reducing sugars and formation of Maillard reaction compounds. 

## 4. Conclusions

The work demonstrates that sucrose addition is an effective strategy to improve *Z. mobilis* leavening performances, increasing microbial growth, as well as gaseous production and retention, with a consequent higher dough development. Even if in the tested conditions the same leavening performances of *S. cerevisiae* were not achieved, results highlight the high potentialities associated with the use of *Z. mobilis* in combination with low sucrose levels (2.5 g/100 g). 

Further studies are required in order to investigate the effects of other ingredients on the leavening performances of *Z. mobilis* DSM 424. In particular, attention will be paid to the application of *Z. mobilis* in doughs with high concentrations of added sugars, salt, and fats; these high osmotic stress conditions are already reported to affect yeast leavening performance [30]. As well as currently happens for *S. cerevisiae*, research should aim at selecting the best performing *Z. mobilis* strain for each kind of dough preparation (i.e., bread, croissant, pizza, traditional festivity Italian cake—Panettone). This diversification would facilitate people with inflammatory bowel disease and/or intolerance reaction to *S. cerevisiae*, which have to avoid *S. cerevisiae* ingestion, excluding most of the traditional baked goods from their diet.

## Figures and Tables

**Figure 1 foods-09-00089-f001:**
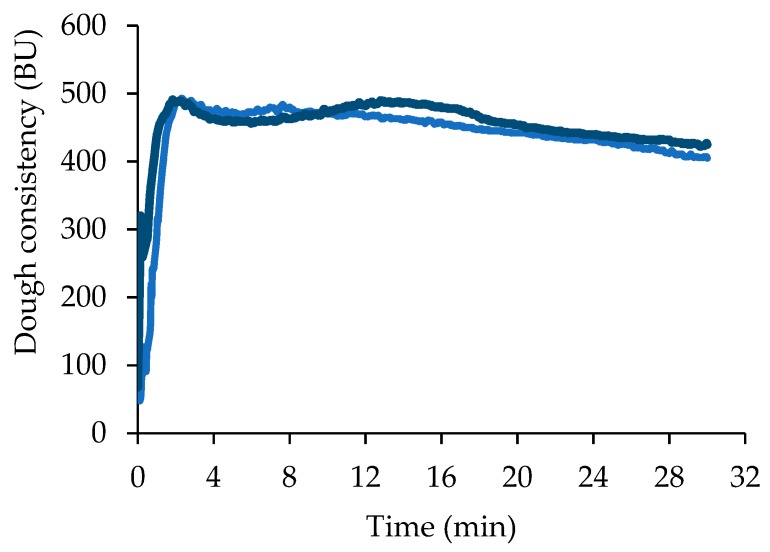
Mixing properties of flour without (light blue) and with (blue) sucrose addition (average values of two replicates, relative standard deviation values ≤10%). BU, Brabender Unit.

**Figure 2 foods-09-00089-f002:**
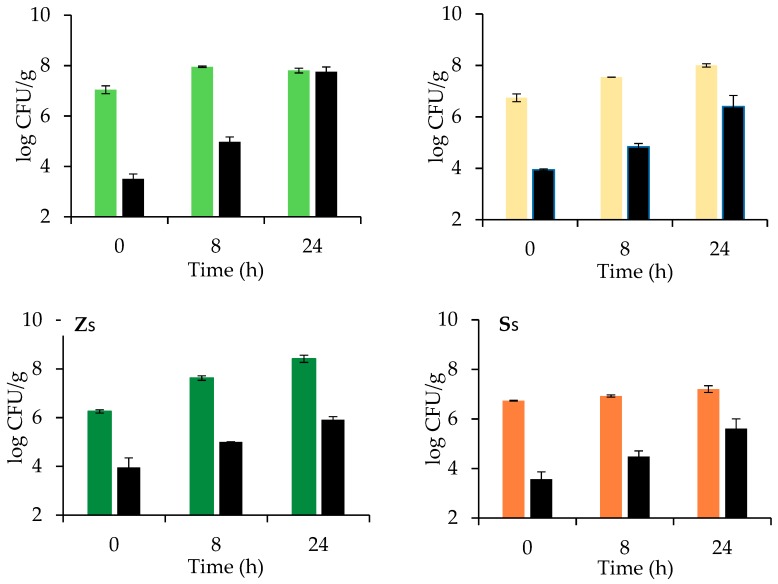
Microbial counts of *Z. mobilis* (green), *S. cerevisiae* (orange) and total bacteria (black) after 0, 8, and 24 h of leavening at 26 °C of doughs fermented by *Z. mobilis* (Z, light green) or *S. cerevisiae* (S, light orange) without and with sucrose (Zs and Ss respectively, green and orange) addition (average values of two technological replicates; error bars represent standard deviation values). See Table 1 for sample identifications and formulations.

**Figure 3 foods-09-00089-f003:**
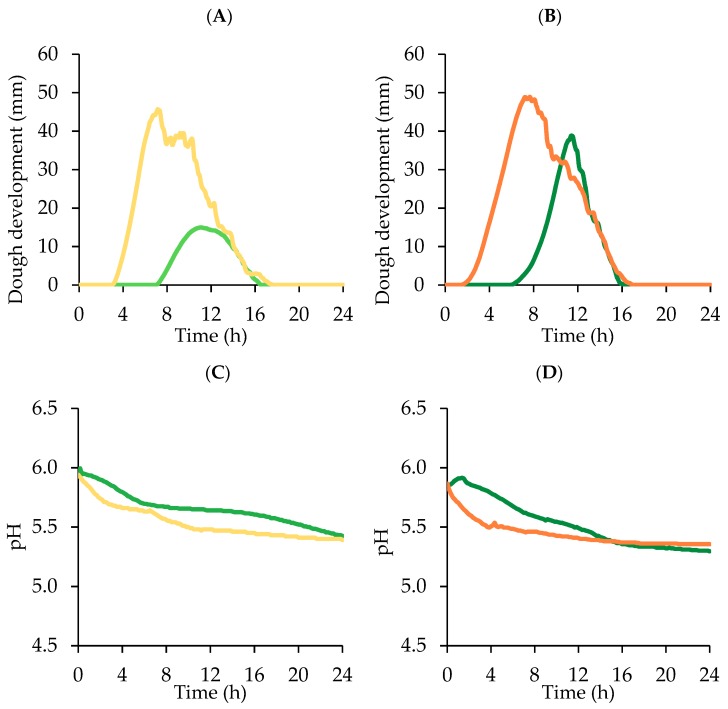
Dough development (**A**,**B**) and pH decrease (**C**,**D**) of doughs leavened for 24 h at 26 °C by *Z. mobilis* (Z, light green) or *S. cerevisiae* (S, light orange) without and with sucrose (Zs and Ss respectively, green and orange) addition (average curves of two technological replicates; relative standard deviation values ≤9%). See Table 1 for sample identifications and formulations.

**Table 1 foods-09-00089-t001:** Dough sample identifications and formulations: ingredients are expressed on flour basis, while inocula on dough basis.

Sample ^1^	Flour (g/100 g)	Water (g/100 g)	Sucrose (g/100 g)	*Z. mobilis* (log CFU/g)	*S. cerevisiae* (log CFU/g)
Z	100	63.3	-	7	-
Zs	100	57.3	2.5	7	-
S	100	63.3	-	-	7
Ss	100	57.3	2.5	-	7

^1^ Z, dough with *Z. mobilis*; Zs, dough with *Z. mobilis* containing sucrose; S, dough with *S. cerevisiae*; Ss, dough with *S. cerevisiae* containing sucrose.

**Table 2 foods-09-00089-t002:** Maltose, glucose, sucrose, and ethanol concentrations (g/100 g flour; mean ± standard deviation values of two technological replicates) in doughs leavened at 26 °C by *Z. mobilis* (Z) or *S. cerevisiae* (S), without and with sucrose (s) addition (2.5 g/100 g flour) at 0, 8, and 24 h of leavening. See Table 1 for sample identifications and formulations.

	Leavening Time (h)	Sample			
		Z	Zs	S	Ss
Maltose	0	1.84 ± 0.38 ^bc^	1.49 ± 0.04 ^ab^	2.09 ± 0.04 ^c^	1.27 ± 0.13 ^a^
	8	3.08 ± 0.10 ^b^	2.36 ± 0.28 ^a^	2.13 ± 0.09 ^a^	2.55 ± 0.11 ^a^
	24	3.85 ± 0.08 ^b^	3.38 ± 0.37 ^b^	n.d.	2.21 ± 0.17 ^a^
Glucose	0	0.27 ± 0.05 ^b^	0.03 ± 0.01 ^a^	0.36 ± 0.01 ^c^	0.62 ± 0.02 ^d^
	8	0.02 ± 0.01 ^a^	0.07 ± 0.03 ^a^	0.09 ± 0.01 ^a^	0.79 ± 0.05 ^b^
	24	n.d.	0.07 ± 0.02 ^a^	n.d.	0.35 ± 0.05 ^b^
Sucrose	0	n.d.	2.78 ± 0.36 ^a^	n.d.	2.05 ± 0.04 ^a^
	8	n.d.	2.76 ± 0.29 ^b^	n.d.	0.17 ± 0.07 ^a^
	24	n.d.	n.d.	n.d.	n.d.
Ethanol	0	n.d.	n.d.	n.d.	n.d.
	8	0.24 ± 0.01 ^a^	0.10 ± 0.07 ^a^	1.18 ± 0.14 ^b^	0.81 ± 0.01 ^b^
	24	0.75 ± 0.01 ^a^	1.72 ± 0.01 ^b^	3.49 ± 0.29 ^d^	2.58 ± 0.05 ^c^

n.d.: not detectable (<0.01 g/100 g flour); ^a–d^ For each compound, different superscript letters within a raw indicate significantly different samples (*p* < 0.05).

**Table 3 foods-09-00089-t003:** Dough leavening properties (mean ± standard deviation values of two technological replicates) of samples leavened for 24 h at 26 °C by *Z. mobilis* (Z) or *S. cerevisiae* (S) with or without sucrose (s) addition. See Table 1 for sample identifications and formulations.

Leavening Indices ^1^	Z	Zs	S	Ss
Hm (mm)	14.95 ± 0.21 ^a^	39.7 ± 1.41 ^b^	45.95 ± 2.33 ^c^	49.45 ± 1.06 ^c^
T1 (h)	11.16 ± 0.06 ^c^	11.55 ± 0.21 ^c^	7.87 ± 0.35 ^b^	7.87 ± 0.35 ^b^
LLT (h)	7.12 ± 0.12 ^d^	6.07 ± 0.02 ^c^	3.19 ± 0.12 ^b^	1.56 ± 0.04 ^a^
LR (mm/h)	5.16 ± 0.05 ^a^	6.86 ± 1.54 ^a^	14.56 ± 0.95 ^c^	10.60 ± 0.10 ^b^
CO_2-TOT_ (mL)	855 ± 136 ^a^	1542 ± 38 ^b^	2231 ± 20 ^c^	2086 ± 136 ^c^
CO_2-RET_ (mL)	739 ± 3 ^a^	1368 ± 23 ^b^	1570 ± 27 ^c^	1574 ± 2 ^c^
Tx (h)	10.75 ± 0.78 ^c^	11.26 ± 0.34 ^c^	6.57 ± 0.07 ^a^	8.29 ± 0.12 ^b^

^a–d^ Values with different superscript letters within a row are significantly different (*p* < 0.05). ^1^ Hm, dough maximum height; T1, time at which the dough maximum height occurs; LLT, lag leavening time; LR, leavening rates; CO_2-TOT_, gaseous production; CO_2-RET_, gaseous retention; Tx, time of dough porosity appearance.

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
