# Peer review of "Zymomonas mobilis in Bread Dough: Characterization of Dough Leavening Performance in Presence of Sucrose"

_foods, 2020, doi:10.3390/foods9010089_

Round 1
Reviewer 1 Report
Please see:
bibliographic references: see lines 36, 95 and 229. et al - line 46 (et al.).
The text becomes more difficult to read given the large amount of systematically and repeatedly placed information in parentheses, forcing the reader to stop numerous times reading and reasoning. Much of this information is already mentioned above.
Has the compressed baker's yeast been evaluated before use? Was it verified if compressed baker's yeast consisted only of Saccharomyces cerevisiae? Was this assessment also made when the compressed yeast was used after being kept at 4 °C within two weeks of purchase and when it could still be used?
Author Response
Dear all,
please find here attached the revised version of the paper Foods-688628, as suggested by the reviewers.
Here follow the answers to the posed questions.
Reviewer 1:
Please see: bibliographic references: see lines 36, 95 and 229. et al - line 46 (et al.).
The revisions were made accordingly.
The text becomes more difficult to read given the large amount of systematically and repeatedly placed information in parentheses, forcing the reader to stop numerous times reading and reasoning. Much of this information is already mentioned above.
Authors have modified the paper accordingly, numerous parentheses have been eliminated and sentences rephrased. All modifications have been made visible using the "Track Changes" function in Microsoft Word.
Has the compressed baker's yeast been evaluated before use? Was it verified if compressed baker's yeast consisted only of Saccharomyces cerevisiae? Was this assessment also made when the compressed yeast was used after being kept at 4 °C within two weeks of purchase and when it could still be used?
Baker’s yeast, commercially available in compressed or dried form, is an industrial microbial preparation containing cells of Saccharomyces cerevisiae. In a previous paper also cited here (“Rollini, M.; Casiraghi, E.; Pagani, M.A.; Manzoni, M. Technological performances of commercial yeast strains (Saccharomyces cerevisiae) in different complex dough formulations. Eur Food Res Technol 2007, 226, 19-24) authors have characterized microbial and technological features of four different commercial baker’s yeasts, five lots each strains, in compressed form. They all contained only S. cerevisiae cells and presented a maximum humidity of 70% (biomass dry weight higher than 30%) according to the Italian legislation limit (75% maximum humidity, 8% maximum ash); total cell counts showed an average of 4 × 1010 cell/g dw.
In the present paper authors have inserted a sentence in the Materials and Methods section to explain how the cell count of compressed baker’s yeast was performed (line 93-97). Moreover, in line 201-205, authors have motivated the choice of using compressed baker’s yeast in the first period of its shelf life, being higher in this period the yeast intracellular glutathione content, an indicator of cell stress tolerance. These results were achieved in a previous research, whose reference has been added (new Ref. 25): Musatti, A.; Manzoni, M.; Rollini, M. Post-fermentative production of glutathione by baker’s yeast (S. cerevisiae) in compressed and dried forms. New Biotechnol 2013, 30, 219-226.
We hope to have fulfilled all requirements for this submission, and are looking forward to hearing from you in due course.
Yours sincerely,
Alida Musatti and Cappa Carola
Reviewer 2 Report
This article was very interesting and I felt it was appropriate to publish it immediately.
Comments to the Author
It is hard for me to see the light gray in the figure.
Foods are an open-access journal.
Since this journal is an open-access journal, I think you can publish color charts for free.
Please change the black and white figures to the color figures.
Question to the author
The author added 2.5g-sucrose/100g-flour.
Why did you choose this condition?
Is it bad to add glucose to the dough instead of adding sucrose to the dough?
Author Response
Dear all,
please find here attached the revised version of the paper Foods-688628, as suggested by the reviewers.
Here follow the answers to the posed questions.
Reviewer 2:
It is hard for me to see the light gray in the figure. Foods are an open-access journal. Since this journal is an open-access journal, I think you can publish color charts for free. Please change the black and white figures to the color figures.
Authors have submitted the new Figure 1, Figure 2 and Figure 3 in colour.
Question to the author:
The author added 2.5g-sucrose/100g-flour. Why did you choose this condition?
Authors have chosen to reduce the amount of sucrose added respect to the data already present in the literature: Oda and Tonomura [12] tested a minimum of 5 g/100 g flour of sucrose up to 35 g/100 g flour. To the best of our knowledge, no data are available concerning sucrose addition lower than 5 g/100 g flour. We have chosen to test a low sucrose addition (2.5 g/100 g flour) to avoid either any sweet modification of the final taste of the leavened product and any modification of its nutritional properties due to the high glycaemic impact. These sentences are already present in lines 66-69 and 72-74.
In order to better clarify the concept, authors have added the following sentence (line 77-80): “Thus, the aim of this work was to investigate the effect of a low sucrose addition (2.5 g/100 g flour) on the dough leavening performance of Z. mobilis, in order to limit any possible modification of the sensory and nutritional features of the final leavened product.”
Is it bad to add glucose to the dough instead of adding sucrose to the dough?
The reviewer is right. Since both S. cerevisiae and Z. mobilis can ferment glucose and no negative effects
to the dough are known due to glucose addition, glucose can be an alternative source of sugar for the bacteria fermentation. Actually, as reported in lines 69-74, this option was already investigated by Musatti, A.; Mapelli, C.; Rollini, M.; Foschino, R.; Picozzi, C. Can Zymomonas mobilis substitute Saccharomyces cerevisiae in cereal dough leavening? Foods 2018 7, 61. doi: 10.3390/foods7040061.
In the present paper the use of sucrose has been investigated, being a cheaper carbon source (263 Euro/10 kg, Sigma) than glucose (336 Euro/10 kg, Sigma): for this reason, if possible bakery companies prefer to use economic ingredients. Authors have added a sentence in line 77-78 commenting also these data.
We hope to have fulfilled all requirements for this submission, and are looking forward to hearing from you in due course.
Yours sincerely,
Alida Musatti and Cappa Carola